# Prevalence and factors associated with comorbid depression and anxiety in patients with diabetes mellitus attending a national referral hospital in Uganda

Sophia Balinga[1,2]*, Blessed Tabitha Aujo[3,4], Wilber Ssembajjwe[5], Paul Bangirana[1], Catherine Abbo[6], Emmanuel Kiiza Mwesiga[1]

1 Department of Psychiatry, College of Health Sciences, Makerere University, Kampala, Uganda,
2 Butabika National Referral Mental Hospital, Kampala, Uganda, 3 Soroti University, Soroti, Uganda,
4 Soroti Regional Referral Hospital, Soroti, Uganda, 5 Medical Research Council/Uganda Virus Research Institute, Kampala Uganda and London School of Hygiene and Tropical Medicine, Kampala, Uganda,
6 University of Ottawa, Department of Psychiatry & Children's Hospital of Eastern Ontario, Ottawa, Canada

* sophiebalinga123@gmail.com

## Abstract

Comorbid depression and anxiety in patients with diabetes mellitus (DM) is associated with poor disease outcomes, yet its burden and associated factors are not well-characterized in low-resource settings like Uganda. This study aimed to determine the prevalence and factors associated with comorbid depression and anxiety among patients with DM in Uganda. A cross-sectional study was conducted at the diabetes clinic of Mulago National Specialised Hospital. The study consecutively enrolled 223 adult patients (≥18 years) with DM. The Mini-International Neuropsychiatric Interview (MINI) was used to diagnose depression and anxiety disorders. Data on socio-demographics, clinical characteristics, alcohol use (AUDIT-C), social support (MSPSS), and quality of life (WHOQOL-BREF) were collected. Logistic regression was used to identify factors associated with comorbid depression and anxiety. The mean age of participants was 54.6 years (SD = 13.1), and 72.2% were female. The prevalence of comorbid depression and anxiety was 14.3% (95% CI: 10.3–19.6). In the multivariable analysis, factors significantly associated with increased odds of comorbidity included having a higher number of children (Adjusted Odds Ratio, aOR=1.15, 95% CI: 1.02–1.32), longer duration since DM diagnosis (aOR=1.07, 95% CI: 1.01–1.12), high blood pressure (aOR=2.19, 95% CI: 1.94–5.08), and moderate/high alcohol use (aOR=1.46, 95% CI: 1.09–5.14). Conversely, diagnosis of Type II DM (aOR=0.40, 95% CI: 0.16 - 0.91), older age (aOR=0.97, 95% CI: 0.94–0.99) and higher scores across all WHOQOL-BREF domains (physical, psychological, social, environmental) were associated with significantly reduced odds of comorbidity. Nearly one in seven patients with DM in this Ugandan cohort had comorbid depression and anxiety. The findings underscore the need for integrated mental health screening and

**Data availability statement:** All relevant data are within the paper and its Supporting information files.

**Funding:** This study was supported by a research scholarship from the African PsyCare Research Organisation (USD 1,331 to SB). The funders had no role in study design, data collection and analysis, decision to publish, or preparation of the manuscript. The content in this manuscript is solely the responsibility of the authors and does not necessarily represent the official views of the APRO.

**Competing interests:** The authors have declared that no competing interests exist.

intervention within diabetic care services, particularly targeting younger patients, those with Type 1 DM, longer disease duration, hypertension, and hazardous alcohol use.

## Introduction

Diabetes Mellitus (DM) is a common and debilitating disorder with increasing prevalence in Low- and Middle-Income Countries (LMIC)), including Uganda, where its prevalence was estimated at 4.6% in 2021 [1]. People with DM have a two-to-threefold increased risk of common mental disorders including depression and anxiety compared to the general population [2]. There is considerable variability in the prevalence of comorbid depression and anxiety among individuals with diabetes mellitus (DM) across Sub-Saharan Africa, ranging from 15.2% to 45.9%. Notably, the prevalence of comorbid major depressive disorder (MDD) in Uganda was 34.8%, significantly higher than in high-income countries such as the Netherlands, where it was reported at 15.3% [3].

While common mental disorders are often studied in isolation, their co-occurrence presents a unique clinical challenge. Comorbid depression and anxiety in DM is associated with greater symptom severity, poorer glycaemic control, reduced quality of life, higher healthcare costs, and increased all-cause mortality [3,4], with diabetic patients experiencing a 1.7 to 2.3 times higher risk of death compared to those without depression [5]. Moreover, depression and anxiety in DM is common among young single females of lower education and socio-economic status, who have medical comorbidities and physical impairment [5].

In LMICs, where mental health services are often fragmented from chronic disease care, most studies have focused solely on depression or relied on screening tools rather than diagnostic interviews, potentially underestimating the unique burden of comorbidity [6]. This study sought to determine the prevalence and associated factors of comorbid depression and anxiety in patients with diabetes mellitus attending Mulago National Specialized hospital (MNSH).

## Methods

### Ethics statement

Ethical approval was obtained from the Makerere University School of Medicine Research and Ethics Committee (Mak-SOMREC-2023–745). Written informed consent was obtained from all participants, with caretakers present when required, after the study was explained in English and Luganda. The study adhered to ethical standards of confidentiality, privacy, and voluntary participation; withdrawal did not affect care. Anonymity was ensured through unique identification numbers, and all data were securely stored with access limited to the principal investigator.

### Study design and setting

A cross-sectional study was conducted at the DM clinic of MNSH in Kampala, Uganda, between 1st February 2024 and 31st March 2024. MNSH is the largest national referral and teaching hospital in Uganda. The weekly diabetes clinic serves over 100 patients.

## Study population and sampling

The study enrolled adult patients (≥18 years) with a confirmed diagnosis of DM for at least three months, who provided written informed consent. Patients requiring urgent medical attention were excluded.

The Kish Leslie (1965) formula [7] was used to estimate sample size using a single proportion of 15.3% reported by Liu et al., (2020) [3], yielding a minimum of 200 participants. This was adjusted upwards by 10% to 223 to account for potential missing data. Participants were consecutively recruited from the daily clinic register until the desired sample size was achieved.

## Data collection and measures

The principal investigator trained psychiatric clinical officers (interviewers) on data collection techniques and clinical assessment using the different scales. Data were collected through face-to-face interviews and medical record abstraction using standardized tools.

1. **Socio-demographic and Clinical Data**: A researcher-administered questionnaire captured data on age, gender, education, occupation, income, and marital status. Clinical data, including DM type, duration, comorbidities, and medications, were abstracted from patient files. Blood pressure and BMI were measured during the visit.

2. **Mental Health Diagnosis**: The Mini-International Neuropsychiatric Interview (M.I.N.I.) version 7.0.2 [8] was used to diagnose Major Depressive Episode and anxiety disorders (Generalized Anxiety Disorder, Panic Disorder, Social Anxiety Disorder, and Agoraphobia). The M.I.N.I. is a validated, structured diagnostic interview compatible with the Diagnostic and Statistical Manual of Mental Disorders (DSM-5) criteria.

3. **Alcohol Use**: The Alcohol Use Disorders Identification Test-Consumption (AUDIT-C) [9] was used to assess alcohol consumption. The AUDIT-C assesses the frequency and quantity of alcohol consumption using three specific questions. Each question is scored from 0 to 4 points, resulting in a total possible score of 0–12. The scores were categorized as low-risk (0–3) or moderate/high-risk (≥4) of alcohol use disorder.

4. **Perceived Social Support**: The Multidimensional Scale of Perceived Social Support (MSPSS) [10] examined perceived social support through a 12-item self-report questionnaire with 7-point Likert scale answers that measure feelings of support from three distinct sources (family, friends, and a significant other). The Likert scale's response descriptors were used guide interpretation of perceived social support. An average Likert score was computed from the 12 items per individual and perceived social support was categorized as low (1-2.9), moderate (3–5), or high (5.1-7) support.

5. **Quality of Life**: The World Health Organization Quality of Life-BREF (WHOQOL-BREF) [11] was used to assess four domains: physical health, psychological health, social relationships, and environment. The total score ranges from 0 to 100 and WHOQOL-BREF in this study was assessed on a continuous scale where higher scores indicate better quality of life.

## Data analysis

All analyses were performed using Stata software (version 18). Descriptive statistics were summarized as frequencies (percentages), means (±standard deviations) or medians (interquartile ranges). The prevalence of comorbid depression and anxiety was reported as a proportion with a 95% confidence interval (CI).

Bivariate logistic regression was used to assess crude associations between independent variables and comorbid depression/anxiety. Variables with a p-value <0.2 in bivariate analysis were included in a multivariable logistic regression model, which was adjusted a priori for age and sex. Crude Odds Ratios (cOR) and Adjusted Odds Ratios (aOR) with 95%

CIs were reported. ORs indicate associations between patient characteristics and comorbid depression or anxiety, but they do not prove cause-and-effect.

A multinomial regression model was fitted to asses relationship with having neither condition, having either depression or anxiety, and having comorbidity. This was done while adjusting for age and sex. Crude relative risk ratios (cRRR) and adjusted relative risk ratios (aRRR) were reported for each explanatory variable. Model development followed a stepwise approach, beginning with demographic factors. Model diagnostics were computed, and statistical significance was set at p-value < 0.05. The ORs describe the "likelihood" which means probability of comorbid depression and anxiety happening.

## Results

A total of 223 participants were enrolled and included in the analysis. The mean age was 54.6 years (SD = 13.1), and the majority were female (161, 72.2%). Over half of the participants had a primary level of education or less (139, 62.4%), and 76.7% had Type 2 DM. The median duration since DM diagnosis was 5 years (IQR: 2–13). More than half (120, 52.8%) had high blood pressure at the time of assessment. The full socio-demographic and clinical characteristics are presented in Table 1.

### Prevalence of comorbid depression and anxiety

The prevalence of comorbid depression and anxiety was 14.3% (32/223; 95% CI: 10.3–19.6). The prevalence of a depressive disorder alone (without a comorbid anxiety disorder) was 36.8% (82/223), and the prevalence of an anxiety disorder alone (without a comorbid depressive disorder) was 22.9% (51/223). Generalized Anxiety Disorder was the most common anxiety disorder (19.7%, 44/223). The comorbidity profile detailed in Fig 1 shows how depression overlaps with different anxiety disorders. The bars indicate the number of participants with a particular disorder or comorbidity represented by the colour codes below them.

### Factors associated with comorbid depression and anxiety

Each additional child was associated with 15% higher odds (aOR=1.15, 95% CI: 1.02-1.32) as shown in Table 2. Participants with Type 2 DM had 60% lower odds than those with Type 1 DM (aOR=0.40, 95% CI: 0.16 - 0.91). Each additional year since diagnosis was associated with 7% higher odds (aOR=1.07, 95% CI: 1.01-1.12). Participants with high blood pressure had 2.19 times higher odds (aOR=2.19, 95% CI: 1.94-5.08). Those with moderate/high-risk alcohol use had 1.46 times higher odds (aOR=1.46, 95% CI: 1.09-5.14). Each year increase in age was associated with a 3% reduction in odds (aOR=0.97, 95% CI: 0.94-0.99). In this cross-sectional study, Quality of Life domains were analyzed as associated variables. While poorer Quality of Life may appear as a predictor of comorbid depression and anxiety, it is equally plausible that it represents a consequence of these conditions. Given the study design, causality cannot be inferred, and the findings should be interpreted as associations only. Higher scores in all four WHOQOL-BREF domains were strongly associated with reduced odds: physical Health (aOR=0.87, 95% CI: 0.80-0.93), psychological Health (aOR=0.80, 95% CI: 0.71-0.89), social relationships (aOR=0.68, 95% CI: 0.58-0.81), environment (aOR=0.85, 95% CI: 0.78-0.92).

Individuals working for pay outside the home had 57% increased likelihood of comorbid depression and anxiety (RRR = 9.64, 95% CI: 1.57 to 59.10, p-value = 0.014) as shown in Table 3. Whereas Quality of Life domains were analyzed as associated variables, it is equally plausible that these represent consequences of comorbid depression and anxiety. Better physical health was associated with 16% lower likelihood (RRR = 0.84, 95% CI: 0.77 to 0.93, p-value = 0.001) of comorbid depression and anxiety while better psychological health was associated with 20% lower likelihood (RRR = 0.80, 95% CI: 0.68 to 0.94, p-value = 0.008) of comorbid depression and anxiety.

**Table 1. Demographic and clinical characteristics of the participants.**

| Factor | Level | Frequency (%) *n = 223* |
|---|---|---|
| Age | Mean (SD) | 54.6 (13.1) |
| Gender | Male | 62 (27.8) |
| | Female | 161 (72.2) |
| Region | North | 9 (4.0) |
| | East | 19 (8.5) |
| | Central | 133 (59.6) |
| | West | 49 (22.0) |
| | Rwandese | 13 (5.8) |
| Religion | Christian | 179 (80.3) |
| | Moslem | 43 (19.3) |
| | Other | 1 (0.4) |
| Marital Status | Single/Never Married | 5 (2.2) |
| | Married/Cohabiting | 115 (51.6) |
| | Separated/Divorced | 63 (28.3) |
| | Widowed | 40 (17.9) |
| Highest Level of Education | No Formal Education | 45 (20.2) |
| | Primary Education | 94 (42.2) |
| | Secondary Education | 56 (25.1) |
| | Post-Secondary School Institution | 40 (12.6) |
| Work for pay outside the home | No | 141 (63.2) |
| | Yes | 82 (36.8) |
| Average Monthly Income | 0 - 100,000 | 121 (54.3) |
| | 100,000 - 500,000 | 59 (26.5) |
| | 500,000 - 1,000,000 | 33 (14.8) |
| | > 1,000,000 | 10 (4.5) |
| Number of children | Median (IQR) | 5 (3– 7) |
| Type of diabetes | Type 1 | 52 (23.3) |
| | Type 2 | 171 (76.7) |
| Duration from time of diagnosis | Median (IQR) | 5 (2–13) |
| Fasting blood sugar level | Median (IQR) | 8.9 (6.8 -11.8) |
| Body mass index | Underweight | 5 (2.2) |
| | Normal | 104 (46.5) |
| | Overweight | 80 (35.9) |
| | Obese | 34 (15.3) |
| Blood Pressure | Normal | 103 (46.2) |
| | High | 120 (52.8) |

IQR: interquartile range, SD: standard deviation.

## Discussion

The prevalence of comorbid depression and anxiety in in patients with diabetes mellitus was high with nearly one in every seven patients showing comorbid depression and anxiety. The observed prevalence is comparable to rates reported in Dutch [3] and Emirati studies [12] which showed 15.3% and 12.5% occurrence of comorbidity respectively but lower than some reports from other regions like India at 21.0% [13]. The high prevalence in this study could be due to poor

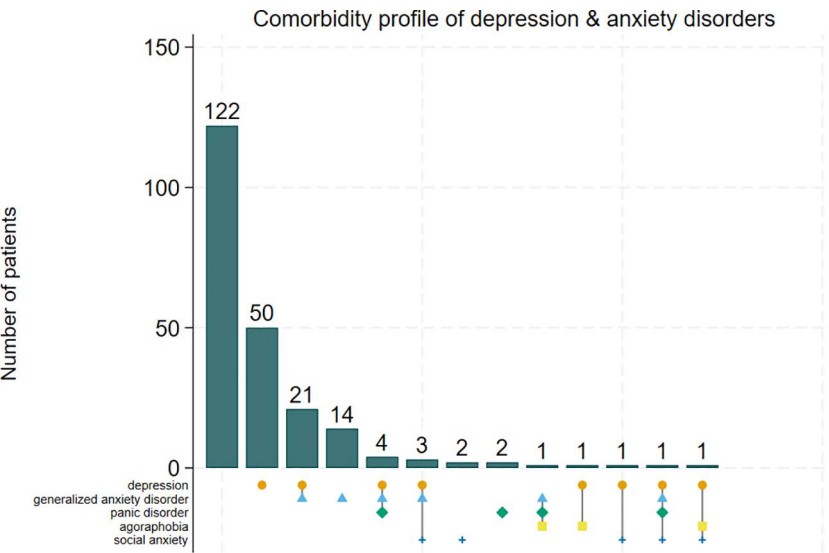

**Fig 1. Comorbidity profile of depression and the different anxiety disorders.**

self-management behaviours seen among patients with diabetes mellitus in Sub-Saharan Africa. This variation may also be due to differences in assessment tools, study populations, and cultural factors influencing the expression of mental distress. Patients in this setting rarely monitor their glucose levels and poorly adhere to recommended dietary and medication behaviour which increase the risk for developing complications from diabetes mellitus hence increasing the risk of developing depression and anxiety [14]. This underscores that comorbid depression and anxiety is a common problem among patients with DM in Ugandan.

There was slight 3% decrease in the likelihood of comorbid depression and anxiety with increasing age. A Malaysian study by [14] reported that older age was associated with a 4% reduced likelihood of occurrence of depression only in DM. On the contrary, a study by Woon et al., [15] reported that increasing age was associated with 43.2% increased risk for comorbid depression and anxiety. The protective effect of increasing age may reflect better coping strategies developed over time or psychological adaptation to living with a chronic illness [16].

Having more children was associated with 4% increased likelihood of comorbid depression and anxiety. This is comparable to a study among women diagnosed with diabetes mellitus, each additional child predicted a 0.282-point increase in anxiety score and a 0.348-point increase in depression score [17]. This result could be due to the financial pressures resulting from having many children, hence making the parents prone to psychological problems like depression and anxiety.

Patients with type-2 diabetes mellitus were 60% less likely to have comorbid depression and anxiety compared to type-1 diabetes mellitus. A narrative review reported that depression only was 26–58% less likely in people with type-2 DM [18]. meta-analysis studies suggest that anxiety disorders and symptoms are 1.2- and 1.5-times more prevalent in people with diabetes compared to the general population respectively [19,20] but these studies do not specify variation within types of DM. The higher co-occurrence of depression and anxiety among type-1 DM might be explained by the higher psychological burden of the demanding regimen of insulin therapy, dietary restrictions, earlier onset of the disease, social stigma and risks of complications [21].

Patients with a longer duration from the time of diagnosis were 7% more likely to have comorbid depression and anxiety per year increase. This is comparable to results in an Egyptian study which reported that anxiety and depression was

**Table 2. Factors associated with comorbid depression and anxiety in diabetes mellitus.**

| Factor | Level | cOR (95% CI) | p-value | aOR (95% CI) | p-value |
|---|---|---|---|---|---|
| Age | Per year increase | 0.97 (0.95- 1.00) | 0.063 | 0.97 (0.94 - 0.99) | **0.047** |
| Gender | Male | 1 | | 1 | |
| | Female | 1.80 (0.70- 4.61) | 0.222 | 1.85 (0.72 - 4.78) | 0.205 |
| Region | Northern | 1 | 0.087 | 1 | 0.200 |
| | Eastern | 0.94 (0.07-11.97) | | 0.88 (0.07 - 11.47) | |
| | Central | 1.50 (0.18- 12.63) | | 1.35 (0.16 - 11.65) | |
| | Western | 0.52 (0.05- 5.66) | | 0.47 (0.04 - 5.20) | |
| | Rwanda | 5.00 (0.47- 52.96) | | 3.48 (0.31 - 38.63) | |
| Religion | Christian | 1 | 0.071 | 1 | 0.105 |
| | Moslem | 2.16 (0.94- 4.99) | | 2.01 (0.86 - 4.71) | |
| | Other | – | | – | |
| Marital Status | Single/Never Married | 1 | 0.549 | 1 | 0.333 |
| | Married/Cohabiting | 0.51 (0.05- 4.92) | | 0.76 (0.07 - 8.84) | |
| | Separated/Divorced | 0.75 (0.08- 7.48) | | 1.02 (0.08 - 12.91) | |
| | Widowed | 1.00 (0.10- 10.22) | | 2.11 (0.14 - 31.60) | |
| Highest Level of Education | No Formal Education | 1 | 0.265 | 1 | 0.286 |
| | Primary Education | 3.32 (0.92- 11.91) | | 3.35 (0.92 - 12.15) | |
| | Secondary Education | 2.33 (0.58- 9.37) | | 2.31 (0.56 -9.45) | |
| | Post-Secondary School | 1.68 (0.31- 8.97) | | 1.89 (0.34 -10.57) | |
| Work for pay outside the home | No | 1 | | | |
| | Yes | 1.89 (0.89- 4.13) | 0.097 | 2.00 (0.88 - 4.56) | 0.099 |
| Average Monthly Income | 0 - 100,000 | 1 | 0.466 | 1 | 0.430 |
| | 100,000 - 500,000 | 0.57 (0.22-1.51) | | 0.58 (0.21 - 1.60) | |
| | 500,000 - 1,000,000 | 1.12 (0.41- 3.07) | | 1.28 (0.45 - 3.61) | |
| | > 1,000,000 | – | | – | |
| Number of children | Per unit increase | 1.04 (0.92- 1.17) | 0.069 | 1.15 (1.02 - 1.32) | **0.004** |
| Type of diabetes | Type 1 | 1 | **0.014** | 1 | **0.012** |
| | Type 2 | 1.30 (1.10-2.83) | | 0.40 (0.16 - 0.91) | |
| Duration from diagnosis | Per year increase | 1.05 (1.00- 1.10) | **0.036** | 1.07 (1.01 - 1.12) | **0.013** |
| Fasting blood sugar level | Per unit increase | 1.06 (0.99- 1.14) | 0.104 | 1.05 (0.98 - 1.14) | 0.166 |
| Body mass index | Underweight | 1 | 0.616 | 1 | 0.740 |
| | Normal | 0.67 (0.07- 6.45) | | 0.80 (0.08 - 8.23) | |
| | Overweight | 0.51 (0.05- 5.05) | | 0.66 (0.60 - 7.15) | |
| | Obese | 1.04 (1.00- 10.81) | | 1.22 (0.11 - 13.89) | |
| Blood Pressure | Normal | 1 | | 1 | |
| | High | 2.77 (1.81-3.88) | 0.151 | 2.19 (1.94 - 5.08) | **0.038** |
| Social support | Low | 1 | 0.054 | 1 | 0.101 |
| | Moderate | 0.60 (0.22- 1.64) | | 0.61 (0.22 - 1.67) | |
| | High | 0.30 (0.11- 0.82) | | 0.33 (0.12 - 0.93) | |
| Alcohol use | Low | 1 | | 1 | |
| | Moderate/high | 1.20 (1.14- 2.60) | 0.051 | 1.46 (1.09 - 5.14) | **0.039** |
| Outcomes of comorbidities | | | | | |
| Quality of life (higher scores = better quality of life) | Physical health | 0.88 (0.82- 0.94) | **< 0.001** | 0.87 (0.80 - 0.93) | **< 0.001** |
| | Psychological | 0.80 (0.72- 0.89) | **< 0.001** | 0.80 (0.71 - 0.89) | **< 0.001** |
| | Social relationships | 0.69 (0.58- 0.82) | **< 0.001** | 0.68 (0.58 - 0.81) | **< 0.001** |
| | Environment | 0.86 (0.79- 0.93) | **< 0.001** | 0.85 (0.78 - 0.92) | **< 0.001** |

aOR: adjusted odds ratios, CI: confidence interval, cOR: crude odds ratios.

**Table 3. Factors associated with either depression or anxiety and comorbidity.**

| Factor | Level | Neither condition | Depression or Anxiety | | Comorbidity | |
|---|---|---|---|---|---|---|
| | | | aRRR (95% CI) | p-value | aRRR (95% CI) | p-value |
| Age | Per year increase | Base outcome | 0.96 (0.91- 1.01) | 0.137 | 0.95 (0.87- 1.04) | 0.293 |
| Gender | Male | Base outcome | 1 | | 1 | |
| | Female | | 0.52 (0.19- 1.43) | 0.212 | 3.79 (0.53- 26.96) | 0.183 |
| Region | North | Base outcome | 1 | 0.543 | 1 | 0.459 |
| | East | | 8.16 (0.40- 165.34) | | 1.69 (0.02- 96.49) | |
| | Central | | 2.55 (0.16- 40.24) | | 1.50 (0.05- 42.25) | |
| | West | | 2.70 (0.16- 45.10) | | 0.10 (0.002- 5.03) | |
| | Rwandese | | 1.04 (0.02- 43.15) | | 7.14 (0.11- 438.19) | |
| Religion | Christian | Base outcome | 1 | 0.324 | 1 | 0.321 |
| | Moslem | | 0.77 (0.26- 2.29) | | 1.02 (0.21- 4.79) | |
| | Other | | – | | – | |
| Marital Status | Single/Never Married | Base outcome | 1 | 0.379 | 1 | 0.471 |
| | Married/Cohabiting | | 0.10 (0.001- 6.90) | | 0.10 (0.001- 6.90) | |
| | Separated/Divorced | | 0.31 (0.004- 19.94) | | 0.31 (0.004- 19.94) | |
| | Widowed | | 2.14 (0.01- 239.15) | | 2.14 (0.01- 239.15) | |
| Highest Level of Education | No Formal Education | Base outcome | 1 | 0.223 | 1e | 0.234 |
| | Primary Education | | 1.18 (0.39- 3.57) | | 3.91 (0.42- 35.75) | |
| | Secondary Education | | 1.05 (0.31- 3.47) | | 2.03 (0.19- 21.47) | |
| | Post-Secondary | | 2.16 (0.47- 9.80) | | 7.26 (0.32- 163.61) | |
| Work for pay outside the home | No | Base outcome | 1 | | 1 | |
| | Yes | | 0.63 (0.21- 1.83) | 0.402 | 9.64 (1.57- 59.10) | **0.014** |
| Number of children | Per unit increase | Base outcome | 0.95 (0.79- 1.13) | 0.639 | 1.08 (0.84- 1.38) | 0.540 |
| Type of diabetes | Type 1 | Base outcome | 1 | 0.403 | 1 | 0.232 |
| | Type 2 | | 1.77 (0.46- 6.76) | | 0.29 (0.04- 2.16) | |
| Duration from diagnosis | Per year increase | Base outcome | 1.02 (0.95- 1.09) | 0.489 | 1.07 (0.95- 1.20) | 0.211 |
| Fasting blood sugar level | Per unit increase | Base outcome | 1.06 (0.96- 1.17) | 0.199 | 1.10 (0.96- 1.25) | 0.135 |
| Body mass index | Underweight | Base outcome | 1 | 0.067 | 1 | 0.065 |
| | Normal | | 0.27 (0.01- 4.58) | | 0.09 (0.002- 3.46) | |
| | Overweight | | 0.37 (0.02- 6.09) | | 0.27 (0.006- 11.49) | |
| | Obese | | 0.68 (0.03- 13.39) | | 0.91 (0.01- 45.65) | |
| Blood Pressure | Normal | Base outcome | 1 | | 1 | |
| | High | | 1.16 (0.51- 2.63) | 0.716 | 2.71 (0.59- 12.34) | 0.196 |
| Social support | Low | Base outcome | 1 | 0.089 | 1 | 0.092 |
| | Moderate | | 1.26 (0.32- 4.83) | | 0.62 (0.08- 4.73) | |
| | High | | 1.07 (0.27- 4.29) | | 1.04 (0.12- 8.50) | |
| Alcohol use | Low | Base outcome | 1 | | 1 | |
| | Moderate/high | | 1.63 (0.15- 16.81) | 0.677 | 0.83 (0.02- 27.46) | 0.918 |
| **Outcomes of comorbidities** | | | | | | |
| Quality of life (higher scores = better quality of life) | Physical health | Base outcome | 0.84 (0.77- 0.93) | **0.001** | 0.90 (0.77- 0.98) | **0.023** |
| | Psychological | Base outcome | 0.80 (0.68- 0.94) | **0.008** | 0.73 (0.58- 0.92) | **0.008** |
| | Social relationships | Base outcome | 0.90 (0.73- 1.10) | 0.326 | 0.67 (0.49- 0.92) | **0.015** |
| | Environment | Base outcome | 1.04 (0.94- 1.16) | 0.360 | 0.91 (0.77- 1.07) | 0.281 |

aRRR: adjusted relative risk ratio, CI: confidence interval, cRRR: crude relative risk ratio.

associated significantly with a longer duration of diabetes mellitus (more than 10 years) [22]. This result could be due to development of complications, treatment burden as the disease progresses. However, this in contrast to results in a Saudi Arabian study which reported that depression and anxiety was 56%-61% less likely among patients having diabetes for more than 10 years [23]. These results suggest that the cumulative burden of disease management, potential development of complications, and diabetes distress over time significantly impact mental health

Patients with high blood pressure during the visit were 2.2 times more likely to have comorbid depression and anxiety. This is comparable to results in various studies which found that depression or anxiety were significantly associated with other comorbidities like high blood pressure [22,24,25]. The link with high blood pressure highlights the clustering of physical and comorbid depression and anxiety. The relationship is likely bidirectional; psychological distress can exacerbate high blood pressure through physiological pathways (such as HPA axis dysregulation) [26] and behavioural mechanisms (like poor adherence to medication, unhealthy diet, and lack of physical activity) [27]. This underscores for an integrated care approach that addresses both high blood pressure and mental health concurrently.

Patients with better physical health were 13% less likely to have comorbid depression and anxiety. Patients with better psychological were 20% less likely to have comorbid depression and anxiety. Patients with better social relationships were 32% less likely to have comorbid depression and anxiety. Patients living in better environment were 15% less likely to have comorbid depression and anxiety. These observations are comparable to a study in India where comorbid depression and anxiety was associated with low WHO-BREF scores across all domains [22]. Likewise, results in a Ugandan study showed a positive association between depression and a poor quality of life in adults with diabetes mellitus [4]. The protective role of better quality of life across all domains, affirmed by the multinomial analysis, is a critical finding. It indicates that interventions which improve a patient's physical well-being, psychological state, social connections, and living environment could be powerful strategies for preventing and mitigating comorbid mental illness in DM.

Patients with moderate to high use of alcohol were 46% more likely to have comorbid depression and anxiety compared to those with low use. This is comparable to results in a Brazilian study which reported a high proportion of high-risk alcohol consumption in diabetes patients with depression and anxiety symptoms [26]. The association between hazardous alcohol use and comorbidity points to the use of alcohol as a potential maladaptive coping mechanism, which can further worsen both diabetic and mental health outcomes. However, a multinational study using different assessment tools for depression and anxiety reported that substance use was not significantly associated with anxiety and depression [22].

Multinomial analysis revealed that those working for pay outside of the home were 9.64 times more likely to develop comorbid depression and anxiety than those with neither of the disorders. A Turkish study reported a similar harmful association at 2.63 times more likelihood of anxiety and depression in participants working in a paid job [28]. This may be due to demands at work coupled with cost of management, complications from DM among many. However, this contrasts with a study done in Iraq which reported that depression and anxiety was 52%-57% less likely among those who were employed [29].

## Strengths and limitations

A key strength of this study is the use of a standardized diagnostic interview (M.I.N.I.) to diagnose depression and anxiety, which reduces misclassification bias common in studies relying solely on screening tools. However, the findings must be interpreted in light of several limitations. The cross-sectional design precludes causal inference. Odds ratios indicate associations between patient characteristics and comorbid depression or anxiety, but they cannot establish causality. This limits interpretation to likelihood rather than cause-and-effect. The study was conducted at a single tertiary referral hospital, which may limit the generalizability of findings to primary care or community settings where DM is managed. Social desirability and recall bias may have influenced self-reported data. Finally, the use of a single fasting blood sugar measurement is a suboptimal indicator of long-term glycaemic control compared to HbA1c.

## Conclusion

This study reveals that comorbid depression and anxiety affects a significant proportion of patients with diabetes in Uganda and is linked to a complex interplay of clinical and psychosocial factors. Notably, the presence of comorbid depression and anxiety was associated with poorer quality of life. The results in this study highlight the need for the routine integration of mental health screening and services into standard diabetes care in Uganda and similar settings. Particular attention should be paid to patients who are younger, have Type 1 DM, a longer disease duration, co-existing hypertension, or engage in hazardous alcohol use. Future research should focus on developing and testing the feasibility and effectiveness of integrated care models that address both the physical and mental health needs of this population.

## Supporting information

**S1 Dataset. Study participants attending the diabetic clinic at Mulago National Specialized Hospital.**
(XLSX)

## Acknowledgments

We acknowledge staff of the department of Psychiatry at Makerere University College of Health Science and the diabetic clinic of Mulago National Specialised Hospital for their guidance and valuable contributions to this study.

## Author contributions

**Conceptualization:** Sophia Balinga.

**Data curation:** Wilber Ssembajjwe.

**Formal analysis:** Wilber Ssembajjwe.

**Investigation:** Sophia Balinga.

**Methodology:** Sophia Balinga.

**Project administration:** Sophia Balinga.

**Resources:** Sophia Balinga.

**Software:** Sophia Balinga.

**Supervision:** Sophia Balinga, Paul Bangirana, Catherine Abbo, Emmanuel Kiiza Mwesiga.

**Validation:** Sophia Balinga, Paul Bangirana, Catherine Abbo, Emmanuel Kiiza Mwesiga.

**Visualization:** Sophia Balinga.

**Writing – original draft:** Sophia Balinga, Paul Bangirana, Catherine Abbo, Emmanuel Kiiza Mwesiga.

**Writing – review & editing:** Sophia Balinga, Blessed Tabitha Aujo, Paul Bangirana, Catherine Abbo, Emmanuel Kiiza Mwesiga.

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
