## [Decision Letter · Decision Letter 0]

26 Jan 2026

PMEN-D-25-00577

Prevalence and factors associated with comorbid depression and anxiety in patients with diabetes mellitus attending a national referral hospital in Uganda

PLOS Mental Health

Dear Dr. BALINGA,

Thank you for submitting your manuscript to PLOS Mental Health. After careful consideration, we feel that it has merit but does not fully meet PLOS Mental Health’s publication criteria as it currently stands. Therefore, we invite you to submit a revised version of the manuscript that addresses the points raised during the review process.

We look forward to receiving your revised manuscript.

Kind regards,

Elias Ghossoub

Academic Editor

PLOS Mental Health

Journal Requirements:

i. Please clarify all sources of financial support for your study. List the grants, grant numbers, and organizations that funded your study, including funding received from your institution. Please note that suppliers of material support, including research materials, should be recognized in the Acknowledgements section rather than in the Financial Disclosure.

ii. State the initials, alongside each funding source, of each author to receive each grant. For example: "This work was supported by the National Institutes of Health (####### to AM; ###### to CJ) and the National Science Foundation (###### to AM)."

iii. State what role the funders took in the study. If the funders had no role in your study, please state: “The funders had no role in study design, data collection and analysis, decision to publish, or preparation of the manuscript.”

iv. If any authors received a salary from any of your funders, please state which authors and which funders.

2. Please ensure that your Ethics Statement is available in its entirety at the beginning of your Methods section, under a subheading 'Ethics Statement'.

3. Please upload separate figure files in .tif or .eps format. Also, remove the figures from your manuscript file but keep the legends.

https://journals.plos.org/mentalhealth/s/figures

https://journals.plos.org/mentalhealth/s/figures#loc-file-requirements

4. We note that your Data Availability Statement is currently as follows: All data used in this study is attached as supplements.

Reviewers' comments:

Reviewer's Responses to Questions

**Comments to the Author**

1. Does this manuscript meet PLOS Mental Health’s publication criteria? Is the manuscript technically sound, and do the data support the conclusions? The manuscript must describe methodologically and ethically rigorous research with conclusions that are appropriately drawn based on the data presented.? Is the manuscript technically sound, and do the data support the conclusions? The manuscript must describe methodologically and ethically rigorous research with conclusions that are appropriately drawn based on the data presented.

Reviewer #1: Yes

Reviewer #2: Yes

2. Has the statistical analysis been performed appropriately and rigorously?

Reviewer #1: Yes

Reviewer #2: Yes

3. Have the authors made all data underlying the findings in their manuscript fully available (please refer to the Data Availability Statement at the start of the manuscript PDF file)?

The PLOS Data policy requires authors to make all data underlying the findings described in their manuscript fully available without restriction, with rare exception. The data should be provided as part of the manuscript or its supporting information, or deposited to a public repository. For example, in addition to summary statistics, the data points behind means, medians and variance measures should be available. If there are restrictions on publicly sharing data—e.g. participant privacy or use of data from a third party—those must be specified.requires authors to make all data underlying the findings described in their manuscript fully available without restriction, with rare exception. The data should be provided as part of the manuscript or its supporting information, or deposited to a public repository. For example, in addition to summary statistics, the data points behind means, medians and variance measures should be available. If there are restrictions on publicly sharing data—e.g. participant privacy or use of data from a third party—those must be specified.

Reviewer #1: Yes

Reviewer #2: Yes

4. Is the manuscript presented in an intelligible fashion and written in standard English?

Reviewer #1: Yes

Reviewer #2: Yes

Reviewer #1: Thank you for the opportunity to review this work.. This is an important topic and the findings are meaningful, overall the paper is well written, but would benefit from some adjustments. I am attaching a separate document with comments which I hope you will find helpful...

Reviewer #2: Research Summary

This manuscript is clearly written and provides an invaluable information on an important theme that integrates non-communicable diseases (NCDs) i.e., diabetes mellitus and mental health especially in a low-resource setting. While research on such issues is still limited in these settings, this paper proves to provide more literature on the topic. The use of structured diagnostic interview (M.I.N.I) rather than the usual screening tools allows for more accurate determination of the true prevalence and hence offers a strengthened internal validity of the findings.

The manuscript however has a number of clarifications that need to be addressed prior to publication. These appear in the diagnostic description, statistical reporting and interpretation of the odds ratios and mix up of results that may leave the reader confused.

Major Issues

Methods

The manuscript clearly states that depression and anxiety were correctly diagnosed using the MINI (v7.0.2) however, it states that “the M.I.N.I diagnoses depression and anxiety using the Patient Health Questionnaire-9 (PHQ-9) and the Generalized Anxiety Disorder-7 (GAD-7)” (line 96), which is inaccurate. Cut off scores are provided for these screening tools however, the MINI does not depend on cut off scores to diagnose the disorders. The results go ahead to dichotomise the anxiety disorders while the GAD-7 does not categorise these but rather only screens for generalised anxiety disorder.

Action required: Clarify the use of the PHQ-9 and the GAD-7 in this study. If they were embedded in the screening, these should be reported independently. Also clarify on the use of cut-off scores. This is essential for validity and reproducibility of the study.

Results

Several inconsistencies appear in the statistical report creating variability in the text and the tables.

i) The abstract (line 34) and the narrative of the results (line 164) report that type I DM is associated with higher odds of comorbidity (aOR 2.48) but tables 2 and 3 report different figures. The reported odds ratios and reference figures in the tables do not match the narrative. There is also inconsistency in the reported odds ratios (crude vs adjusted) in the abstract and the results.

ii) Extremely wide confidence intervals appear in table 3 which raises concerns on the model stability.

Actions

i) Ensure all reference categories are clearly stated.

ii) Clarify the reported estimated and harmonise these throughout the text (abstract and results).

iii) Recheck the regression models.

Discussion

i) When considering blood pressure, this was categorised as “high and normal” in tables 2 and 3, which was measured during the visit (line 90). In the discussion section (line 232), this is discussed as “Patients with hypertension” which shows an inconsistency in the reporting.

ii) There is repeated interpretation of odds ratios as “increased likelihood” (line 208), “less likely” (line 241) which imply causal inference yet this cannot be ascertained from a cross-sectional study. Using words like “more associated with” would be more appropriate.

Actions:

i) Clarify whether individuals with high blood pressure or hypertension were considered given that the two are not synonymous.

ii) Clarify the association of the factors with the outcome.

Minor Issues

Methods: Interviewer training is not mentioned. It would be more informative to include it.

Results: The tables would benefit from captions. Figure 1 would also benefit from a caption explaining the colour codes.

Discussion: Quality of Life domains are discussed as predictors however, these can also be a consequence of the comorbidity of depression and anxiety. This would benefit from a clarification on this.

**Do you want your identity to be public for this peer review?** For information about this choice, including consent withdrawal, please see our Privacy Policy..

Reviewer #1: No

Reviewer #2: **Yes:** Sadat KatamaSadat KatamaSadat KatamaSadat Katama

---

## [Editor Report · Decision Letter 1]

1 Apr 2026

Prevalence and factors associated with comorbid depression and anxiety in patients with diabetes mellitus attending a national referral hospital in Uganda

PMEN-D-25-00577R1

Dear Dr BALINGA,

We are pleased to inform you that your manuscript 'Prevalence and factors associated with comorbid depression and anxiety in patients with diabetes mellitus attending a national referral hospital in Uganda' has been provisionally accepted for publication in PLOS Mental Health.

Best regards,

Elias Ghossoub

Academic Editor

PLOS Mental Health